# Persistent Enterovirus Infection: Little Deletions, Long Infections

**DOI:** 10.3390/vaccines10050770

**Published:** 2022-05-12

**Authors:** Nora M. Chapman

**Affiliations:** Department of Pathology & Microbiology, University of Nebraska Medical Center, Omaha, NE 68198, USA; nchapman@unmc.edu

**Keywords:** enterovirus, terminal deletion, coxsackievirus B, persistent infection, positive-strand initiation, negative-strand initiation, cis-acting replication element

## Abstract

Enteroviruses have now been shown to persist in cell cultures and in vivo by a novel mechanism involving the deletion of varying amounts of the 5′ terminal genomic region termed domain I (also known as the cloverleaf). Molecular clones of coxsackievirus B3 (CVB3) genomes with 5′ terminal deletions (TD) of varying length allow the study of these mutant populations, which are able to replicate in the complete absence of wildtype virus genomes. The study of TD enteroviruses has revealed numerous significant differences from canonical enteroviral biology. The deletions appear and become the dominant population when an enterovirus replicates in quiescent cell populations, but can also occur if one of the cis-acting replication elements of the genome (CRE-2C) is artificially mutated in the element’s stem and loop structures. This review discusses how the TD genomes arise, how they interact with the host, and their effects on host biology.

## 1. Introduction

“...when you have eliminated all which is impossible, then whatever remains, however improbable, must be the truth.” (Casebook of Sherlock Holmes; Sir Arthur Conan Doyle).

Human enteroviral infections are generally well understood because this cytoplasmically replicating group of viruses can be readily studied in cell culture in great detail [1]. In an acute infection in vivo, the virus replicates cytolytically, generating maximal levels of excretion (via a fecal–oral route of transmission) to infect the next individual [2]. While generally considered to be acute infections that are quickly limited, then ablated, by the host immune response, studies of diseases such as enteroviral cardiomyopathies and type 1 diabetes have, in contrast, often demonstrated the presence of enterovirus or enterovirus RNA in the diseased heart [3] and in pancreatic islets [4] at a stage later than the acute infection, when the host adaptive immune response has been activated. As enteroviruses replicate using only viral RNA as the genetic material and cause lysis of the transfected cell, it has been difficult to understand how the latency of viral RNA alone, in the absence of an infectious virus, and cell survival after acute infection could occur.

It has long been established in cell culture that under certain circumstances, enteroviruses can persist well after the initial acute phase of infection, by carrier cultures [5,6] or defective interfering (DI) genome populations [7]. Persistence of enteroviruses in vivo has also been noted by observing long-term shedding of an infectious virus, an occurrence found to be due to viral replication in immunologically suppressed hosts [8]. We and others have long noted numerous incidences of long-term detection of group B coxsackievirus (CVB) RNA in heart tissue of experimentally infected mice in the apparent absence of an infectious, which is to say cytopathic, virus from the same samples. The proposed hypothetical mechanisms employed to account for these findings were logically unsatisfactory. We therefore set out to determine the mechanism that would account for the detection of enteroviral RNA in the absence of cytopathic virions.

## 2. Characteristics of Persistent Enterovirus Infections

The association of enteroviruses, in particular the six serotypes of the group B coxsackieviruses (CVB1-6), with inflammatory heart disease has been known since the 1950s [9,10]. Other than systemic neonatal infections [11], the diseases that we associate with nonpolio enteroviruses are usually acute and quite diverse [2], but two notable diseases that occur after the induction of the adaptive immune response stand out: dilated cardiomyopathy (DCM) [12] and type 1 diabetes (T1D) [4,13]. The association of enteroviruses with these diseases has been discovered via the detection of viral antigens and RNA with slot blots [14,15,16], immunohistochemistry (IHC) [17], in situ hybridization (ISH) [18,19], and, more recently, with much more sensitive versions of reverse transcription polymerase chain reaction (RT-PCR) [20,21,22]. Early work demonstrated that enterovirus RNA was present in human hearts at a stage when the enterovirus could not be detected from the same tissue using cytolytic assays [23]. Other studies have demonstrated the persistence of viral RNA in mice susceptible to CVB3-induced myocarditis up to 77 to 90 days post-inoculation (p.i.) [24,25], and in other experimental picornavirus infections in mice [26,27,28,29,30].

As enteroviruses were previously understood to be RNA viruses without a true mechanism for latency, it was argued that enterovirus-associated heart disease generates an acute infection, via cytolysis, and induces inflammatory immune responses, which subsequently lead to the development of chronic heart disease [31]. Later work investigating experimental CVB3 infection in mice described an acute infection that was followed by both innate and adaptive immune responses, which cleared the virus (undetectable cytopathic effect (cpe) when tissue lysates were assayed in cell culture) and also generated an autoimmune response against uninfected cardiomyocytes [32]. It was proposed that this would lead to post-viral pathologic changes, which could either persist or gradually fade without the requirement for a persistent enterovirus presence. Examination of CVB3-induced myocarditis in mice indicated that early acute disease was accompanied by the presence of the virus and the infiltration of tissue with high levels of lymphoid cells, while later stages, in which fibrosis and remodeling of the heart occurred, showed no indication of virus presence following cytopathic assays in cell culture. This led to the hypothesis that enterovirus infection could provoke a pathologic autoimmunity that leads to late-stage disease [31,32].

### 2.1. Persistent Infections Can Occur Due to Host Immunodeficiencies

Classically, persistent infection due to immunodeficiency has been studied in individuals with agammaglobulinemia who, following an enterovirus infection, continue to excrete enteroviruses over periods of years [33]. Some long-term excretors of poliovirus (PV) have also been found in studies of PV in vaccinated communities [8].

### 2.2. Carrier Cultures Produce Viruses in the Long Term

Carrier cultures [34] consist of a mixture of virus and cells. There are typically a limited number of cells in the culture that are susceptible to infection at any time, and the virus evolves either to infect more cells or to persist in the culture until susceptible cells are present, while the cell population evolves to resist infection. This mechanism may be related to the availability of receptor molecules on the cell surface and the stability of the virus in culture media [35]. Eventually, there is a fluctuating population of cells and of virions, with the persistence of both in the closed culture. Although studies using carrier cultures can demonstrate mutations that may adapt the virus to a particular cell type and/or changes in the cells in the culture in adaption to the virus [36,37], this does not seem characteristic of in vivo infections of the heart or pancreas, in which the tissue is well differentiated and not dividing. In particular, persistent infections within the heart demonstrate an enterovirus persisting in the well-differentiated, nondividing cardiomyocytes [18,38].

### 2.3. Defective Interfering Viruses Have Genomic Deletions

One mechanism by which enteroviruses can persist in cell culture has been shown to be the production of defective interfering (DI) viruses, defective genomes generated in cells from wildtype viruses that interfere with the replication of the wildtype virus but require the presence of the wildtype genome to replicate [39,40]. These viruses are often generated following high multiplicity of infection (MOI) passages in cell culture [7], but have been found in vivo for a number of different viruses [41,42,43,44]. Enteroviral DI genomes are characterized by deletions within the capsid coding (P1) region [40], and were detected because the deletions altered the density of the defective viruses in cesium gradients [45]. DI viruses replicate RNA more rapidly than wildtype viruses due to shorter DI RNA lengths, but cannot form infectious virion structures without wildtype genomes to provide the missing capsid proteins in trans. The DI viral genomes interfere in replication via competition with the wildtype virus [46]. Although well documented in in vitro systems [40], the production in vivo of the classical defective interfering enteroviruses has not been described.

## 3. Discovery and Characterization of a Novel Late-Stage “Noninfectious” Persistent Enterovirus Mechanism

### 3.1. Structure of the Enterovirus Genome

An enterovirus is a picornavirus [1], i.e., a small, encapsidated (but not enveloped) positive-strand RNA virus. The genomic RNA contains a long open reading frame (ORF) encoding the viral capsid proteins and the proteins required for protein processing and viral RNA replication [1]. There are two nontranslated regions flanking the ORF that contain essential RNA elements important for RNA replication and translation. The 5′ nontranslated region (5′NTR) contains several domains, including regions important for RNA replication, such as the OriL [47], and translation, such as the internal ribosome entry site (IRES) [48]. The 3′NTR (OriR) is followed by a polyA tract. The ORF also contains a cis-acting replication element (CRE-2C or OriI), an RNA structure in the region encoding a viral protein, 2C, in enteroviruses A, B, C, and D [49].

### 3.2. Persistent Noncytopathic Enteroviruses in Hearts Are Infectious

In previous studies on coxsackievirus B3 (CVB3) infections in mice, we observed that HeLa cell culture of heart homogenates from 18–21 days post-inoculation (p.i.) did not result in cytopathology, although similar cultures of heart homogenates from shorter p.i. times demonstrated both cpe and positive detection of viral RNA, detected by RT-PCR (Figure 1) [50]. Virus RNA was observed to persist in mouse hearts for at least 53 days p.i. (longer times were not assayed). Despite the lack of cpe induced in HeLa cells by heart homogenates at longer p.i. times, when anti-CVB3 neutralizing polyclonal antiserum was added to these lysates, detection of CVB3 RNA by RT-PCR was ablated [50]; this finding indicates that the enterovirus detected by RT-PCR was actually infectious and encapsidated.

### 3.3. Persistent CVB Populations in Heart Tissue Demonstrated Deletions at the 5′ Genomic RNA Terminus

To determine the mechanism by which persisting CVB3 was replicating an encapsidated, infectious, but noncytopathic, virus, we examined the persisting CVB3 RNA, with the understanding that we would encounter classical DI-type enteroviral genomes with deletions in the P1 region of the genome. Despite repeated attempts, we were unable to demonstrate any deletions within the viral RNA ORF; instead, we discovered a very unusual and quite unexpected form of deletion: the extreme 5′ end of the positive strand of these genomes was not detectable using RT-PCR [50]. This was an astounding finding as, hitherto, an intact 5′ terminal sequence of the enteroviral genome was believed to be required for replication [51]. It is also worth noting that this may have escaped previous discovery because the 5′ genomic terminus is rarely assayed due to the comparative insensitivity of PCR-based assays utilizing tailed cDNA or rapid amplification of cDNA ends (RACE) [52]. This finding later justified the examination of persisting viral RNA in human myocarditis and cardiomyopathic hearts to determine if TD RNA was present [53,54,55].

### 3.4. The 5′ Terminal Deletions Occur within the 5′ RNA Structural Domain Termed the Cloverleaf

The cloning and complete sequencing of numerous enterovirus genomes demonstrated that the 5′ NTR was conserved to a greater degree than most regions of the ORF. The well-studied 5′ NTR of enteroviruses has been shown to contain six discrete domains, termed I–VI [56,57]. The RNA sequence of domain I of the positive strand (and the RNA of the corresponding region of the negative strand at the 3′ end) folds to form a stem (a) and three stem-loop structures (b, c, and d), which together are often generically termed the “cloverleaf” [47,56,58]. Studies using CVB3 demonstrated that deletions within the 5′ 90 nucleotides produced lethal genomes (ablating the cpe when assayed in cell culture) [59,60], but interestingly, the naturally occurring TD mutations that permitted CVB persistence were all within the cloverleaf of the positive-strand RNA 5′ NTR (Figure 2) [12,50,53,54,61]. As the cloverleaf is both a highly conserved structural region and one in which mutations can limit replication or ablate production of infectious virus [59,60,62], it is therefore not surprising that CVB populations that carry the 5′ terminal deletions (TD) on the genome are noncytopathic in cell culture. While deletions of varying length were observed, the majority of CVB-TD genomes characterized in murine hearts and pancreas [50,63], in primary cell culture passage [61], and in human hearts and peripheral blood cells [12,53,54] had deleted stem a and stem-loop b of the cloverleaf (Figure 2). The longest naturally occurring 5′ terminal deletion that we observed was 49 nucleotides in length [50]. The virus generated from a molecular clone of this RNA (CVB3-TD50) retained most of the structure of stem-loop d [50], a region important for 3CD binding [64,65,66]. The naming of the TD genomes in this review uses the CVB3 genome nucleotide number (GenBank AY752944) of the 5′ terminal nucleotide of the TD genome (for example, TD8 has CVB3 nucleotide 8 as the 5′ nucleotide), as in our first publication of the TDs [50]. Where TDs from other authors are discussed in this review, the name of the TD is altered to this system for consistency, in this discussion, with our studies.

### 3.5. Replication of CVB3-TD Genomes: Roles of Regions of the Cloverleaf

In order to examine CVB3-TD replication, CVB3 genomes with deletions found in persisting infections were cloned in plasmids with an upstream T7 RNA polymerase promoter [50]. These clones incorporated a ribozyme [51] at the 5′ end of the T7 transcript to cleave the transcribed RNA at the authentic 5′ end of the TD genome. Virus populations were raised following transfection of HeLa cell cultures with T7-transcribed positive-strand RNA [50].

Short deletions of fewer than five nucleotides from the 5′ terminus have been shown to revert to wildtype [67]; however, deletions of greater size are present in persisting infections, and do not revert when transfected and passed in HeLa cell culture [50,61]. For this review, wildtype refers to full-length enteroviruses without 5′ terminal deletions. The TD genome with a seven-nucleotide 5′ terminal deletion (CVB3-TD8) lacks most of stem a of the cloverleaf but is infectious, noncytopathic (as are all TD strains), and produces virus particles [50]. Mutations in PV RNA that eliminated the base pairing of stem a, or altered the sequence of nucleotides 3–8 of the PV genome, were reported to have lost the ability to form plaque on HeLa cells [62]. This corresponds to other results, suggesting that the 5′ six-nucleotide sequence is essential for replication, as mutations greatly reduced positive-strand replication in vitro [68].

However, serial passage of CVB3-TD8 (which had the 5′ nucleotides 1–7 of wildtype CVB3 RNA deleted) in primary murine pancreatic and cardiac cells [61] or in mice [50] led to TD virus populations with deletions greater than seven nucleotides. How might this have occurred? When CVB3-TD8 positive-strand genomes are transfected and replicate, they should generate a negative-strand RNA with a seven-nucleotide deletion at the 3′ end. However, if the deleted region is required for positive-strand initiation at the 3′ end of the negative strand, then there must be a mode of positive-strand initiation without this specificity. Initiations on the remaining 3′ end region, but not directed exclusively to the 3′ end, would thus generate a TD genome population with varied deletions. Nonetheless, accumulation of longer 5′ terminal deletions appears to have limits, as passage of the CVB3-TD virus with a terminal 49-nucleotide deletion (CVB3-TD50) in primary cell cultures and in mice failed to create a population of TDs with deletions greater than 49 nucleotides [50,61]. Despite our observation that naturally occurring 5′ TD deletions larger than 49 nucleotides do not occur (or, more likely, are exceedingly rare), we created a CVB3-TD genome deleted through stem-loop d (CVB3-TD78) that retained the ability to replicate in HeLa cells and generate the virus, albeit with a 100,000-fold reduction in virus particle production [69]. We suggest that the lack of observed stable populations of TD CVB genomes with deletions greater than 49 nucleotides from the 5′ end is more likely to be the result of reduced fitness within the maelstrom of constantly occurring TD RNAs in an infected cell, rather than TDs without stem-loop d ever occurring.

Ourselves and others observed losses of part or all of the complete stem-loop b in the majority of enterovirus TDs occurring in primary cell culture, mice, and human heart and peripheral blood [12,50,53,54,55,61]. This cloverleaf domain contains a binding site for polyC-binding protein 2 (PCBP2), which plays a role both in translation and in negative-strand replication [47,70]. There is also a C-rich sequence downstream of the cloverleaf (approximately at nucleotides 90–100 in CVB3), which also binds PCBP2 [71,72,73], and remains present in the TDs generated in vivo. A loss of negative-strand synthesis was observed only in mutants of both regions [71]. Using in vitro translation and replication assays, Lévêque and colleagues demonstrated a difference between CVB3-TD8 and -TD22, and CVB3-TD31 and -TD50, showing that the CVB3-TD8 and -TD22 had greater levels of in vitro translation [74]. While they were unable to demonstrate negative-strand replication with CVB3-TD31 and -TD50 (although the shorter deletions did replicate), an earlier study of CVB3-TD50 did demonstrate both translation in vitro and negative-strand replication using extended exposure times [71]. Despite the loss of the PCBP2 binding site in stem-loop b, the majority of CVB3-TDs replicate in cell culture and produce viral proteins and negative-strand RNA in vitro [74]. The loss of the PCBP2 site may be compensated by the function of the C-rich sequence [71], which, being located downstream of the cloverleaf, is maintained in all TD genomes.

The cloverleaf stem-loop d binds viral protein 3C and its precursors [75], and is predicted to form part of a critical replication complex containing viral protein 3CD [47,49,62,64,68]. The mutation or deletion of nucleotides 48 and 49 in the CVB3 cloverleaf has been shown to lower positive-strand RNA replication in vitro [71]. CVB3-TD50 lacks nucleotides 46–49 in stem-loop d, yet remains viable [50,61,76,77]. Not having found deletions longer than 49 nucleotides, we infer that the retention of the majority of the stem-loop d region indicates that loss of this region substantially reduces virus replication to a level of extinction in vivo. Although (our artificially created) CVB3-TD78 does replicate in permissive HeLa cells [69], it is possible that it would not replicate sufficiently well to compete with shorter deletions in restrictive environments, such as the cardiomyocyte. Taken together, these results suggest that the increasing length of the 5′ genomic terminal deletions in passage [50,61] is generated as a consequence of the stochastic loss of initiation sites. However, the stem-loop d deletion likely results in seriously deleterious effects upon translation and replication, which results in positive selection for those members of the TD genome population that retain it.

### 3.6. Evidence That 5′ Terminal Deletions (TDs) Produce Viruses That Replicate at Very Low Levels

HeLa cultures transfected with wildtype CVB3 genome RNA transcripts produced purified virus yields that are 100,000 greater than CVB3-TD RNA transcripts [76]. When we measured CVB3-TD translation levels in cell cultures, they were markedly lower than wildtype virus levels, but nonetheless detectable, although demonstration of VP1 expression required at least 26 h p.i. (whereas 5–6 h is normal for wildtype viruses, which normally begin to lyse HeLa cells by 8 h) [50]. In contrast, several laboratories reported transfecting CVB3-TD genomes in HeLa, HL-1 cells, and human cardiac myocytes without producing evidence of RNA replication shortly after transfection [12,59,74]. Clearly this vastly lowered RNA replication rate requires a longer incubation of cultures as well as longer exposure time of the detection assay to see a signal. In our work to detect CVB3-TD RNA in cardiac homogenates and cell cultures inoculated with infected cardiac homogenates, we used RT-PCR with primers specific for the 5′-NTR. For the examination of the replication of individual terminal deletions, we transfected HeLa cells in bulk, cultured them for 72 h, and concentrated the HeLa lysates through sucrose gradients 50–100 fold. The progeny virus from these large-scale transfections demonstrated that the virus was encapsidated and had the same equilibrium density in CsCl isopycnic gradients as wildtype CVB3 [50]. It is necessary to be specific about methods as the lack of detection with other methods has often been mistakenly understood as an inability of the CVB3-TD viruses to replicate.

Hunziker and colleagues [59] demonstrated no plaque formation upon transfection of CVB3-TD33 RNA in HeLa cells, nor production of virus and anti-CVB3 immunity in mice transfected with the viral RNA. In addition, these authors reported less viral protein generated in CVB3-TD33-transfected HeLa cells than in wildtype-transfected cells and no progeny viral RNA generated (as demonstrated by slot blot of cellular RNA from 0 to 24 h post-transfection). These authors concluded that deletions of nucleotides 1–32 of the CVB3 genome do not give rise to infectious progeny [59]. Despite failing to detect viral RNA replication, host cell protein translation shutoff in HeLa cells transfected with TD33 was evident [59], demonstrating an effect of the CVB3 viral proteases even at the low levels of translation and replication. In our own work, we never observed cpe by inoculating HeLa cells with any population of purified CVB3-TD virions–although RNA was demonstrable by RT-PCR [50,54,61]. The extent of cytopathogenicity of the TD strains was insufficient to be observed by plaque assay; when infected cells finally lysed, surrounding uninfected cells rapidly filled in the space. Using CVB3-TD8 or -TD50 viral preparations, we inoculated mice intraperitoneally with purified virus amounts equivalent to 5 × 10^4^ positive-strand RNA copies. As far as day 175 p.i., virus RNA was detectable in heart tissue by RT-PCR [50]. The serum from these mice contained binding antibodies, which bound the parental CVB3, but neutralizing antibodies were not demonstrable (N.M. Chapman, S. Tracy, unpublished data). Doubtless, the inoculation of mice with an encapsidated virus carrying CVB3-TD deletions, rather than using an inoculation of naked RNA delivered into muscle [59], results in persistent murine infection with the CVB3-TDs, but the neutralizing antibody was not generated in either case.

Using wildtype and CVB3-TD replicons expressing luciferase, Lévêque and colleagues [74] demonstrated that transfected cardiomyocytes at 6 h produced 50–150-fold less luciferase with CVB3-TD replicons than wildtype CVB3 in cardiomyocytes. Treatment with guanidine hydrochloride, an inhibitor of negative-strand replication, did not differentiate between treated and untreated CVB3-TD-transfected cells after 8 h, suggesting that no significant RNA replication had occurred during this time. In vitro translation showed lower viral protein levels when conducted using RNA from larger TDs (CVB3-TD31 and -TD50) than with RNA from shorter deletions (CVB3-TD8 or CVB3-TD22); moreover, in vitro replication assays demonstrated both double-stranded and single-stranded RNA products with TD8 and TD22, but at a fraction of the levels of the wildtype. As discussed above, Sharma et al. [71] found evidence of RNA replication, but only after an extended exposure. Bouin and colleagues [12] demonstrated the ability of TD-transfected RNA to produce sufficient 2A protease in order to cleave eIFG4, despite the low level of viral protein translation. Cumulatively, these findings are similar to the reported in vitro results with CVB3-TD33 [59] and, again, demonstrate that transfection (which results in productive infection in only part of the transfected cells) is not sufficient to measure replication within hours of transfection or without concentration of the progeny virus [50,76]. Techniques sufficiently sensitive to detect replication with the wildtype virus are not necessarily sufficiently sensitive for studies of CVB3-TD with severely hindered replication capacity.

It is clear that the loss of major parts of the cloverleaf structure of the CVB genome, which reduces replication by an extreme amount, is not a complete barrier to replication. Although translation assays in cell-free assays demonstrated that the CVB3-TD50 had a 50% reduction in translation, the reduction in positive-strand replication was comparatively greater [74]. This is not unexpected; we demonstrated that CVB3-TD50 produces only 1/100,000 of the yield of wildtype encapsidated CVB3 in HeLa cells [76]. However, even with a greatly diminished level of replication, the CVB3-TD50 virus could be passaged in cell culture and used successfully to infect mice [50,61].

At this point, it is important to stress that careful work has demonstrated that CVB3-TD populations can exist without the presence of any wildtype (or helper) virus. CVB3-TD virion preparations were initiated by transfection with RNA transcribed from molecular clones of CVB3-TD genomes and, thus, no wildtype virus RNA was present [50]. Furthermore, extended incubation and serial passages of such CVB3-TD transfections produced no cpe, which would have occurred had any wildtype virions been present [50,77]. Therefore, because the CVB3-TDs tested in HeLa cells produced an encapsidated virus without the presence of helper/wildtype RNA, there is evidently sufficient translation to produce the viral proteins and the virus. Analysis of mutations in the cloverleaf have examined the effects of mutations with transfected cells and in vitro assays (e.g., [12,59,74]). Virus populations with 5′ terminally deleted genomes were initially discovered in vivo, and were assayed by passage in cells in culture that involved the isolation of the encapsidated virus to confirm that the RNA was from viral replication [50,61,76]. It is apparent that infection by normal viral pathways is more efficient than using RNA transfection with these viruses.

Given that TD genome populations arise in tissues following acute wildtype virus infections, it seems likely that the process by which TDs arise moves at some rate within the infected cell, and thence to other cells, from the initial infecting virus genome to mixed populations of wildtype virus and TD genomes, to populations dominated by TDs, and eventually to tissues in which only TD RNA populations are present. In a study on the multiple passage of wildtype CVB3 in primary cell cultures [61], we noted that virus yield titers (by cpe) decreased in the second and third passages to undetectable levels by the third or fourth passage (Figure 3). Using an RT-PCR with a primer that targeted nucleotides 1–20, we could detect 5′ terminal sequences, but by the third or fourth passages no wildtype (that is, intact) 5′ termini were detectable, despite all passages having detectable CVB3 RNA using standard RT-PCR [61]. As the cpe of these passages (indicating a wildtype virus) was present but decreased from passage 1, it can be assumed that the second and third passages contained either fewer infectious wildtype virus genomes and/or a mixture of wildtype CVB3 and CVB3-TD RNA. As the 5′ terminal primer (S) [50] used for RT-PCR detection of the 5′ terminal sequences would detect any 5′ sequences with deletions of less than 10 nucleotides, the positive RT-PCR with this assay for passages 2 and 3 also indicated residual wildtype CVB3 in these early passages, supporting the hypothesis of gradual increases in TD genomes in tissues with persistent enterovirus infections. In addition, three studies have demonstrated that enteroviral TDs were accompanied by a minor wildtype population in cardiac tissue and peripheral blood samples from patients with DCM and myocarditis, using a very sensitive assay [12,53,55]. Despite having clearly shown that CVB3-TDs can replicate in mice and cell culture without any wildtype virus [50,54,61,76,77], these observations suggest that there may nonetheless be a selective advantage to having a minor population of wildtype virus in the cardiomyocyte enterovirus quasispecies. Alternatively, and we deem this more likely, in those tissues assayed by Bouin and colleagues, the shift in quasispecies from wildtype to TD genome may not have been completed at time of sampling. Although a co-infection by another enterovirus during the course of the development might provide intact 5′ termini, the very low level of the wildtype 5′ termini would not be characteristic of an acute infection [50,61,63,76,77].

### 3.7. The Replication of CVB3-TD Strains Is Similar to Wildtype CVB3 in That the Initiation Process Produces RNA Covalently Linked to VPg

As the picornaviruses use a protein primer, VPg and its precursors, to initiate the positive and negative strand [49], the question arose as to whether the mechanism of initiation of the TD viral RNA also involved the covalent link of VPg to the viral RNA. This was an obvious question, for all TD strains had deleted 5′ termini and, in general, most CVB3-TD genomes studied did not possess uridine residues at the 5′ positive-strand RNA terminus (Figure 2). Desiring to examine newly replicated TD viral RNA and, as only newly replicated enterovirus RNA is encapsidated [78], we purified TD virions from culture by treatment of cleared cell lysates with RNase, concentrating the virions by centrifugation through sucrose gradients and banding in CsCl density gradients [50]. To test whether VPg was covalently attached to TD RNA, viral RNA was isolated with Trizol from CsCl-banded virions and used for a Western blot of the RNA probed with an antiserum to a PV VPg peptide [50]. This work demonstrated that VPg covalently attached to viral RNA from wildtype PV and CVB3 (positive controls) as well as from CVB3-TD8 and -TD50 [50]. Equal amounts of viral RNA were used for all of the viruses run on the gel, and both the wildtype and CVB3-TD RNAs had an equivalent amount of VPg (Figure 4). This result indicates that although the CVB3-TD genomes have 5′ terminal deletions with different 5′ terminal dinucleotides, the process of initiation still includes the use of the protein primer, VPg.

### 3.8. Persistent, Nonlytic CVB3-TD Populations Demonstrated a Significantly Altered Positive to Negative RNA Strand Ratio

In picornaviral replication, negative-strand RNA is transcribed from the infecting positive strand and, subsequently, serves as a template for many progeny positive-strand copies; additionally, it is complexed with positive strands in a largely double-stranded replication intermediate (RI) form [47,49]. There are always a larger number of positive-strand RNA copies to negative-strand copies in wildtype enterovirus-infected cells, although the ratio has not been unequivocally decided, varying from 40–70 [79] to 20 [80].

Prior studies of persistent CVB infections have indicated a characteristic decrease in the normally high positive- to negative-strand ratios [26,81,82]. This was confirmed in further studies in biopsies of enterovirus-infected hearts from patients with DCM in which positive- to negative-strand ratios of 2–20 were found [22], as well as in the murine models of enterovirus infection [81,82,83]. The reason for these disparities from cell culture studies was not determined.

Assays for double-stranded RNA in purified RNA preparations from persistently infected tissues can produce a double-stranded form by hybridization in vitro, although the double-stranded replicative form (RF) [84] may be the dominant cellular form to persist in vivo. Mutations in the cloverleaf of the enterovirus 5′ NTR tested with in vitro assays have been shown to alter the positive- to negative-strand ratio [71]. As efficient wildtype positive-strand RNA replication in cell culture also produces large positive- to negative-strand ratios, a decrease in this ratio observed in persistent infections of tissues is an indication that the positive-strand RNA replication is relatively impaired in TD virus populations. It seems likely that the cloverleaf deletions of CVB3-TD viruses could account for the altered positive- to negative-strand ratio in persisting enterovirus infections.

### 3.9. CVB3 TD Infections Encapsidate Both Negative and Positive Strands

Careful work has shown that wildtype PV and CVB3 virions contain no detectable negative-strand RNA [50,79]. Due to the low levels of TD enterovirus production in vivo, it is likely that situations in which enteroviral proteins or RNA are detected in tissues or cells, though not in a form that produces observable cpe, are the consequence of low-level persistent production of the TD genome virus. As encapsidated enteroviral RNA is newly replicated [78] and, consequently, has covalently attached VPg, as shown for both wildtype CVB3 and CVB3-TD [50], there was an expectation that the encapsidated RNA would be all positive strand. Slot blot analyses and strand-specific RT-PCR of RNA isolated from highly purified (RNase-treated, pelleted through sucrose gradient, and CsCl-banded) CVB3 TD virions (Figure 5) demonstrated, however, that the encapsidated CVB3-TD RNA population contained nearly equivalent amounts of positive-strand and negative-strand RNA (ratio of 3:1) [50]. Again, the wildtype CVB3 virion did not encapsidate the negative strand.

These results beg the question: why are only positive strands encapsidated in wildtype enteroviruses, while replication-defective TD enteroviruses can, and do, package both strands? A potential reason for this is that the wildtype enteroviruses have a very efficient positive-strand initiation in comparison to negative-strand initiation, producing multiple single-stranded positive strands generated from an RI containing a single negative strand [49]. This progeny RNA will be positive strand. The deletion of part of the cloverleaf (a site at which mutations have been demonstrated to lower the positive- to negative-strand ratio [71]) and the decrease in positive- to negative-strand ratio of persisting enteroviruses, suggests that TD enteroviruses do not efficiently generate RI, but instead, single-stranded progeny RNA is generated from either site of initiation (3′ end of negative strands or 3′ end of positive strands). If, as is believed, packaging of picornavirus RNA is linked to replication [78,85], altered initiation will induce the encapsidation of either strand. No other mutation of enteroviruses has been found that will generate negative-strand encapsidation; however, given the very low level of replication of the TD genomes, such mutations would have, in retrospect, been considered “lethal”, and not studied at the depth that we present here, for this naturally occurring mutation.

These results [50] also clearly demonstrate that there is no positive-strand-specific RNA packaging signal (reviewed [86]). In more recent work [87], multiple RNA sites have been proposed as RNA packaging signals based on evolutionary conservation, but, given what we know, it is hard to conceive that these would be selected in the noninfectious negative strand.

## 4. Generation of TD Genomes Is Likely Due to a Defect in Positive-Strand Initiation

### 4.1. Wildtype Positive-Strand Replication

It seems evident that the 5′ terminal deletions in the TD virus positive-strand genome are a consequence of altered positive-strand initiation. In addition to the loss of the normal 5′ terminal sequence, the 5′ nucleotides of CVB-TD virus genomes (although linked to VPg) do not favor uridine (Figure 2), indicating that the initiation of these genomes is not dependent on the canonical VPgpUpU primer. Present models of enterovirus RNA replication show that the translation of the positive-strand enterovirus genome generates viral proteins, and that these proteins associate with host factors, the RNA positive-strand cloverleaf, and the 3′ end of the genome to induce the circularization of the RNA and initiation of negative-strand replication [47,49,88]. The association of the enteroviral RNA-dependent RNA polymerase, 3Dpol, with the cis-acting replication complex (CRE-2C) and viral protein 3CD generates the uridylation of VPg and its precursors [89]; these complexes then allow the initiation of the negative-strand RNA at the polyA region of the 3′ end [47,49]. After elongation produces the double-stranded replication form (RF), the association of viral protein 2C and the host protein heterogeneous nuclear ribonucleoprotein C (hnRNP-C) with the positive-strand 3′ end of the negative RNA strand probably opens that end of the RF, allowing specific initiation and preserving the normal 5′ terminal sequence, working with high efficiency so that a replication intermediate (RI) producing many positive-strand copies occurs [47,49,88].

### 4.2. CVB3-TD Populations Arise and Become Dominant in Quiescent Primary Cell Cultures and in Nondividing Tissue In Vivo

Persistent enterovirus infections have been found in cardiomyocytes of adult hearts [12,23,53,54], a tissue noted for its lack of dividing cells [90]. Other tissues providing evidence of persistent enterovirus infection in humans are the pancreatic islets of adult T1D patients [17]. Studies of pancreatic islet cells of recent-onset T1D patients demonstrated that enterovirus capsid protein VP1 was detected only in cells negative for the cellular proliferation marker Ki67 [91]. Although inoculation of neonatal mice with CVB3 demonstrated infection with CVB3 in proliferating neuronal progenitor cells, adult mice (inoculated with CVB3 as neonates) showed persistent low-level infection (noncytopathic but replicating when tested in HeLa cells) with nearly equal positive- to negative-strand ratios, a finding consistent with TD persistence in the adult CNS [83]. Early results for intracranial inoculation in mice using a mouse-adapted strain of PV also provide evidence of enterovirus persistence consistent with the generation of a TD population in the CNS [26]. Use of serum starvation induce put HeLa cultures into a quiescent state prior to infection with CVB3 reduces translation and replication of the virus, and resumption occurs after the cell cycle block is relieved [92]. When infected HeLa cells are held for more than 2 days in a quiescent state, neither the expression of CVB3-encoded green fluorescent protein nor the generation of plaque-forming units was observed when cells were reactivated to cycling by the addition of fetal bovine serum. Importantly, RT-PCR was able to detect a very low level of viral RNA [92]. While these data are suggestive of a persistence in quiescent HeLa cells via a TD mechanism, this experiment should be tested on a greater scale to determine whether this condition simply arrests viral replication or indeed generates an environment in which TD virus replication is induced. Our work on the passage of CVB3 in primary cell cultures suggests that repeated passage in cell-cycle-arrested cultures will increase the frequency of 5′ terminal deletions in the passaged virus [61]. If this block imposed on cell cycling can also prevent normal enterovirus replication, it is possible that cells that have left the cell cycle (G0) may have entered a state in which replication of the enterovirus is occurring without the host cell factors required for normal replication. If so, treatment to produce cell cycle arrest would provide a convenient cell culture system for the examination of enteroviral TD replication. 

### 4.3. Host Cell Factors Binding to the 3′ Nucleotides of the Negative Strand May Play a Critical Role in Localization of the Positive-Strand RNA Initiation

Some host factors are predominantly nuclear in nondividing cells, present only at low levels in the cytoplasm, in which the replication of enteroviruses occurs until the enteroviral infection and activity of the viral 2A protease upon nucleoporins allows nuclear-to-cytoplasmic transport [93,94,95,96,97,98,99]. Enteroviruses have evolved to use these proteins for enhanced viral translation and replication [70,100]. However, the question arises as to how efficient enteroviral translation and replication can occur prior to alteration of the nuclear pore complex in quiescent cells. As shown for hnRNP-C, some cellular states, such as mitosis, can enhance the relocalization of such factors [101,102]. When enteroviruses infect nondividing cells, there may well be less efficient translation of the viral genome, resulting in defective viral replication.

We hypothesize that the primary condition leading to the altered initiation of positive strands, which ends up generating TDs, is a relative lack of essential host factors in nondividing and/or quiescent cells. If suboptimal initiation results in (mistaken) priming of positive-strand RNA not specifically targeted to the 3′ end of the negative strand, these positive-strand RNAs will have deletions of the normal 5′ end sequence, and will then serve as templates for negative strands, which in turn will lack the wildtype negative strand 3′ end. This process multiplies the production of TD RNA, and as these negative strands missing 3′ terminal nucleotides serve as templates for positive-strand RNA initiation, the priming of positive strands will result in more TD RNAs being generated. This needs to be better understood, and a focus of future research should be to examine the effects of enterovirus replication in nondividing and/or quiescent cells, with the goal of identifying the proteins whose absence triggers altered positive-strand initiation.

### 4.4. Heterogeneous Nuclear Ribonucleoprotein-C (hnRNP-C) Is a Likely Candidate for a Host Factor in Positive-Strand Initiation

hnRNP-C is a host protein with a nuclear retention signal that is rarely present in the cytoplasm of actively dividing cells [103]. It is a splicing regulator [104] and a factor in mitotic translation [101,105]. hnRNP-C1 was identified by Semler and colleagues as part of a complex binding to the 3′ end of the negative strand of PV, an efficiency of binding that is greatly reduced by the deletion of the sequence that is the reverse complement of nucleotides 5–10 in the positive strand [106,107]. This deletion in the transfected positive strand prevents viral RNA expression of the transfected viral RNA within 15 h. As hnRNP-C1 has also been shown to bind to both the 5′ and 3′ end of PV negative-strand RNA [108], and deletions of the 3′NTR also have reduced viral RNA replication [108], it is possible that the ability of hnRNP-C to multimerize [109] allows the circularization of the negative strand via this interaction [108]. Mutation of the fourth and fifth nucleotides from the 3′ end of the CVB3 negative strand from UU to CC was also shown to greatly decrease hnRNP-C binding [110]. hnRNP-C interacts with the PV P3 nonstructural proteins as well as viral RNA, suggesting that these complexes might be involved in replication [107]. The fact that 5′ terminal deletions that decrease the binding of hnRNP-C reduce viral replication in vitro and in vivo to varying degrees, and that reduced cytoplasmic expression of hnRNP-C decreases positive-strand PV replication [94,107,108], confirms that hnRNP-C is a host factor required for efficient replication of enteroviruses. 

The presence of hnRNP-C increases the efficiency of positive-strand replication [94], while treatment of cells with miRNA able to silence hnRNP-C translation decreases both cytoplasmic hnRNP-C and the number of PV plaques [111]. hnRNP-C1 binds to 3′ terminal nucleotides of the negative strand (nt 5–10) [106], and thus 3′ terminal deletions are likely to result in the omission, or greatly reduced presence, of this factor from positive-strand initiation complexes bound to the 3′ end of the negative strand. If the role of hnRNP-C1 in initiation is the anchoring of positive-strand initiation of the 3′ end of the negative strand, initiation without it may occur in the context of the 3′ cloverleaf of the negative strand but lack specificity for the 3′ end. This should generate 5′ terminal deletions of the positive-strand progeny. At the time of writing, there is no evidence that a lack of hnRNP-C causes nonspecific initiation.

## 5. A Cis-Acting Replication Element (CRE-2C) Plays a Role in Localization of the Positive-Strand Initiation to the 3′ End of the Negative Strand

### 5.1. CVB3-TD Viruses Are Generated by Replication of CVB3 with Multiple Mutations of the CRE-2C

As CVB3-TD virion RNA was shown to have covalently attached VPg, despite the fact that not all TD genomes have uridine as the 5′ nucleotide [50], the question arose as to whether the CRE-2C, which functions in the uridylation of VPg (reviewed by Paul and Wimmer [49]), is required for the replication of the TD CVB genomes. An inhibitor of VPg binding to the viral 3D polymerase, amiloride, decreases the initiation of enterovirus genome replication of both strands [112], an observation that verifies the process of nucleotidylation of VPg as being a part of the RNA initiation process. It has been shown that the disruption of the CVB3 CRE-2C structure via 16 mutations in the stem and loop (Figure 6; mutations that did not alter the 2C amino acid sequence) inhibited the production of a cytopathic virus in cell culture, and generated double-stranded RF, but not single-stranded viral RNA, when replicons containing these mutations were transfected in cell culture [113]. Similar findings were obtained with an eight-mutation disruption of the PV CRE-2C [114]. These two studies suggested that the lack of generation of single-stranded viral RNA with an altered CRE was due to the decrease in positive-strand RNA replication, although sufficient negative-strand replication occurred for the generation of a double-stranded RF [114,115]. The extent to which negative-strand initiation is dependent upon the generation of uridylated VPg via the CRE-2C has yet to be resolved [47,49,116].

To examine the function of the CRE-2C in CVB3-TD replication, we created an otherwise wildtype CVB3 infectious cDNA clone (with an intact 5′ NTR) using the same 16 mutations (as reported previously to be lethal [113]) in the CRE-2C (termed CVB3-CKO) [76]. We also placed the same mutated CRE-2C in a genome with the 49 nucleotide 5′ terminal deletion (termed CVB3-TD50-CKO). The study of CVB3-CKO and CVB3-TD50-CKO in cell culture showed both viruses replicated at the same diminished level (relative to parental CVB3) observed for CVB3-TD50 (consisting of a 49 nucleotide 5′ deletion and an intact CRE). Compared to wildtype CVB3 replication, purified CVB3-CKO virus replicated to a 100,000-fold lower titer [76]. As these mutated RNA genomes all replicated to produce an encapsidated progeny virus (although at greatly reduced levels compared to wildtype genomes) following RNA electroporation into HeLa cells, it was possible to examine the viral genomes using RT-PCR and sequencing. Indeed, transfection of mice with transcripts of wildtype CVB3 and CVB3-CKO genomes (complexed with a lipid-based in vivo transfection agent and inoculated intraperitoneally) resulted in viral replication, and RNA was detected in hearts and spleens by day 8 (wildtype) and day 20 (CVB3-CKO) [76]. Just as with the CVB3-TD50 and parental CVB3 genomes, purified virus (obtained by electroporating CVB3-CKO RNA in HeLa cells or from spleens of mice previously injected intraperitoneally with lipid-complexed viral RNA) could infect HeLa cells and replicate. This replication was prevented by the presence of anti-CVB3 neutralizing antibody, thus demonstrating that the purified viruses were encapsidated and infectious. Upon examination of the 5′ termini of the CVB3-CKO viral RNA from purified virus stocks, loss of the wildtype 5′ terminal sequence was observed despite the fact that the electroporated CVB3-CKO RNA had contained an intact wildtype 5′ end (cleaved from transcribed genomes by an encoded hammerhead ribozyme) [76].

These experiments demonstrate that the complete disruption of the CRE-2C structure by these mutations was not lethal, as had been reported for enteroviruses [113,117,118], and that instead, replication led to 5′ terminally deleted genomes with the low-level virus yields seen in other CVB3-TD strains [76]. Furthermore, like other CVB3-TD viruses, these CVB3-TD CKO strains did not induce cpe in cell culture assays [76]. Studies of the enterovirus CRE-2C showed that mutation of 8 and 16 nucleotides throughout the CRE stem and loop do not revert in passage in cell culture even when present in a CRE translocated outside of the 2C coding region [113,117,118]. These assays of reversion were cpe-based. Because we now know that transfection of the CVB3-CKO leads to detectable 5′ terminally deleted positive-strand RNA within 5 days in cell culture [77], any assay using 6–10 h post-transfection culture of a multiply mutated CRE that does not revert upon culture or passage in cultures, may well have generated TD viruses that do not induce cpe. As the assay we used for detection of the 5′ terminal sequences detects any 5′ sequences with deletions of less than 10 nucleotides [77], the population of viral RNA at day 1 and 3 post-HeLa electroporation of CKO may have been a mixed population of TD and transfected wildtype genomes as replication proceeds, but by day 5 no wildtype 5′ ends were detected [77]. An earlier study at day 8 post-electroporation using purified virus demonstrated no wildtype 5′ ends in these progeny viruses [76]. Because of this rapid change in the cellular population of viral RNA, we now understand that the function of CRE-2C is clearly involved in the initiation of wildtype positive strands in enteroviruses. How initiation occurs without the CRE-2C, though, is unclear, although it might be due to a lack of efficient VPg uridylation, or to the lack of involvement of the CRE-2C in the initiation complex. 

### 5.2. Reversion of the Mutated CRE-2C to Wildtype in CVB3-TD Genomes Indicates That the Functions of This Structure Are Required for TD Genome Replication

To determine the timing of the generation of the 5′ terminal deletion in the CVB3-CKO virus, electroporation of HeLa cells with CVB3-CKO and CVB3-TD50-CKO RNA was carried out and then assayed over time for 5′ end sequence and the state of the CRE-2C mutations [77]. Mice were also transfected with these RNAs as described using intraperitoneal inoculation [77]. At day 1 and 3, the nested RT-PCR (using a tagged primer targeted to the 5′ most 9 nucleotides of the wildtype CVB3 genome) could still amplify a sequence from the virus RNA. However, at day 5 post-electroporation, no wildtype 5′ sequence could be amplified from RNA in viruses from CVB3-CKO electroporated HeLa. The sequence of the CRE-2C region in viral RNA at day 1, 3, and 5 post-electroporation still demonstrated a population majority for the mutations of the CRE-2C. However, by day 8 in HeLa cells, and day 20 in mice, the complete wildtype CRE-2C had been regenerated [77].

It may appear odd that the complete reversion of the CRE-encoding region was generated by day 8 without detection of intermediate reversions, but it must be remembered that even single-base mutations in the CRE-2C are known to be significantly less functional than in the wildtype sequence [117]. As genomes stochastically accumulate reversions to a semi-functional structure, they are also able to exert a trans-dominant inhibition of replication of other genomes in the cellular population [119]. We hypothesize that only when (a) multiple CRE-2C reversions have been generated, (b) are assembled by recombination, and (c) when the reverted genomes are the majority of the population, would (d) the virus population have the ability to replicate even at the low level of TD viruses. Verification of this hypothesis would require large cultures of transfected cells coupled with isolation of progeny genomes away from transfected viral RNA. We used the isolation of an encapsidated virion to isolate progeny genomes; however, purification of progeny genomes away from transfected viral RNA using anti-VPg antibody is possible [115].

Nevertheless, transfection of the CVB3-CKO or the CVB3-TD50-CKO transcripts produced populations of a virus with 5′ terminal deletions of the genomes, and both also produced a virus with reversion of the CKO mutations to wildtype. The selection of wildtype CRE sequence as the overwhelming majority sequence even in a population that has the 5′ terminal deletions (from electroporation of CVB3-TD50-CKO) is strong evidence that the CRE-2C plays an essential role in CVB3 replication, whether or not the virus is TD. The function of the CRE-2C provides a strong selective advantage, despite the fact that the 5′ termini of the TD genomes display no selection for uridine [50,54,61], notably demonstrating no specific requirement for uridylated VPg in positive-strand TD initiation. 

Is this a demonstration that wildtype positive-strand RNA initiation requires the CRE-2C? Clearly, there is a high requirement for either the uridylation of VPg on the CRE-2C, or the CRE structure itself, to have the targeted initiation of the positive strand on the 3′ wildtype end of the negative strand. It must be noted that although the CRE-2C reverted by day 8 in HeLa cells electroporated with the CVB3-CKO RNA, the reversion did not revive a wildtype genome with intact wildtype 5′ termini [77]. This suggests that in 5 days of replication without a functional CRE-2C, the positive-strand initiation was not directed specifically to the 3′ end of the negative strand, but occurred within the cloverleaf, generating 5′ terminal deletions. Had it been otherwise, our assays with RT-PCR for the intact 5′ end would have detected the presence of wildtype 5′ ends in the purified progeny virus at day 5 [77]. Passage of the progeny virus would also allow the escape of any non-TD viruses with a reverted CRE because they would replicate to much greater levels than any TD virus, and would have appeared as a cytolytic population on HeLa cell monolayers [77]. Assays of electroporation of mixed populations of wildtype CVB3 and CVB3-CKO RNA generated cpe in 1 day when as few as 100 copies of wildtype CVB3 were electroporated, together with 2 × 10^11^ copies of CVB3-CKO [77]. These findings cumulatively suggest a very strong effect upon positive-strand RNA initiation in vivo by the mutation of CRE-2C.

Is the reversion of the CRE-2C a demonstration that wildtype negative-strand initiation requires the CRE-2C? It should be considered that initiation of the negative strand at the 3′ end of the positive strand might be enhanced by the uridylation of VPg, because inhibition of VPgpUpU production has been suggested to inhibit negative-strand initiation [112]. There is in vitro evidence that mutation of the enteroviral CRE-2C still allows negative-strand synthesis; sequence analysis indicates that negative strands are initiated when using genomes with a nonfunctional CRE-2C [115]. The production of an RF without the production of an RI or single-stranded positive-strand viral RNA [114] does indicate production of negative-strand RNA, and our work indicates that nonfunctional mutations of the CRE-2C still replicate an infectious virus [76,77]. However, mutations in the CVB3 CRE-2C that reduce uridylation of VPg, also reduce the amount of RF produced in vitro [113]. This indicates that while initiation of the negative strand without a CRE-2C can occur, the efficiency of this process is enhanced sufficiently by a functional CRE-2C to select reversion at all mutated sites in the quasispecies. The argument that there is an alternate, but less efficient, mechanism for the addition of nucleotides to VPg during initiation, as seen in 3D assays with polyadenylated RNAs [120], suggests that VPg could function as a primer without uridylation on the negative-strand RNA; however, the reversion in the CVB3-TD populations (which cannot use the uridylated VPg as a primer) suggests that initiation of the negative strand is much more efficient with a functional CRE-2C.

As there is a use for the multiply mutated CRE-2C as a lethal mutant, together with RNA carrying the intact CRE sequence in recombination assays for enteroviruses [121], this assay should be used with the knowledge that transfection of the genome with the multiply mutated CRE-2C may well generate a replication-competent genome with 5′ terminal deletions within a few days of replication, and the CRE-2C may regenerate shortly thereafter. As such an assay uses a large amount of transfected RNA of two genomes, one of which has a functional CRE, the success of the assay in producing recombinant progeny indicates that the action in trans of the CRE in the transfected acceptor RNA allows normal positive-strand initiation on the negative-strand copy of the transfected donor RNA with the mutated CRE-2C. Copy choice recombination of the elongating positive strands to the donor positive strand could then create sufficient wildtype genomes to prevent any TD virus generation. The fact that these assays produce recombinants provides another example of trans activity of the CRE-2C originally demonstrated in PV [115].

This work indicates strongly that the CRE-2C has an important role in generating wildtype positive-strand initiation. The generation of TDs in quiescent or nondividing cells in vivo is neither so rapid nor so complete as the effect of the CKO mutation on the induction of TD genomes. This role could be a requirement for VPg uridylation and/or the interaction of the CRE-2C with complexed proteins in the wildtype positive-strand initiation complex.

## 6. Enteroviral TDs as Defective Interfering RNA?

### 6.1. TD Viruses Are Produced in Populations That Include Wildtype Virus

Our study that produced TD virus populations by passaging parental (wildtype) CVB3 in cultures of quiescent primary cells derived from murine heart and pancreas, and from human cardiac fibroblasts, relied on taking the viral progeny of an infection of these cells and passaging that preparation on a fresh monolayer of primary cells, then repeating the process [61]. By the third or fourth passage, the progeny virus was no longer inducing cpe when assayed on HeLa cells, but the presence of virus in these infected HeLa cells was shown by RT-PCR and by the production of infectious virus. One explanation of the requirement for repeated passages of the wildtype in quiescent primary cell cultures to produce CVB3-TD populations is that the infection of the primary cells produces a mixed population of parental (wildtype) and TD virus. On each passage, the replication of the TD virus, although low, continued, while the parental virus continued to produce a mixed population of wildtype and TD. On successive passages, the ratio of wildtype to TD RNA decreased until the wildtype virus was no longer detectable [61]. Again, as our assay for the presence of wildtype 5′ termini would detect very low levels of undeleted viral genomes in a large background of TDs, it is clear that the TD viruses accumulated despite the presence of wildtype in early passages [50,61].

### 6.2. The Presence of TD Virus in a Mixed Population Lowers the Level of Replication of the Wildtype Genomes

Primary human cardiomyocyte cultures transfected with CVB3-TD50 or CVB3-TD51 RNA did not produce replicating enterovirus at a level that was detectable by luciferase assays of replicons (comparing cells treated with guanidine and without) after 8 h [74], or by quantitative RT-PCR [12]. Previous work [50,69,76,77] that aimed to produce CVB3-TD by transfection always used transfection of HeLa cells with a specific CVB-TD positive-strand (genomic) RNA generated from cloned genomes, an approach which produced a very low level of virus RNA, detectable by RT-PCR, and viral proteins only after long incubation times [50,69,76,77]. This virus could then be purified and concentrated. Production of CVB3-TD virus by passage [61] involved infection of cells with encapsidated virus and infection of mice with purified concentrated encapsidated virus. A very low level of replication in cardiomyocyte cells by wildtype CVB3, which in our case required more than 48 h to exhibit an increase in yield [122], could have lowered the level of replication of the TD viruses below detectable levels in the transfections. As TD viruses replicate very poorly, even in the highly permissive environment of HeLa cells, expression in cardiomyocytes may require highly extended times for detectable replication. Although transfection allows the insertion of greater amounts of RNA into cells, infection using purified concentrated virus may be more efficient at producing productive infection per viral RNA. This is not applicable, however, to wildtype viruses (or mutant viruses with substantial ability to replicate), as productive transfer of even a minimum number of genomes will rapidly replicate virus and rapidly infect other cells. In our work on CVB3-TD infection of HeLa cells, we showed detectable expression of VP1 (using Western blot) and detectable replication (using RT-PCR) [50,76,77]. In addition, CVB3-TD8 and CVB3-TD50 could be passaged productively upon murine cardiomyocytes [61]. The examination of replication of these CVB3-TDs in human cardiomyocytes by infection (using stocks of virus prepared by transfection and concentration of virus from HeLa [50,76,77]) may overcome the difficulties of transfection. It has to be remembered that although these viruses replicate very poorly, they are a naturally occurring consequence of enterovirus infection, as shown in these studies of human DCM [12,53].

Although it was not possible to show replication of CVB3-TD51 in human cardiomyocytes transfected with RNA, the presence of CVB3-TD51 RNA (19:1, TD51:wildtype) significantly slowed the replication of viral RNA by 24 h compared to replication of wildtype CVB3 alone [12]. Consistent with interference, this mixture also reduced the level of the viral capsid protein VP1 produced in the mixed cultures [12]. This reduction in replication of the wildtype CVB3 suggests that the CVB3-TDs may have the ability to interfere with the replication and translation of the wildtype virus. Studies of enterovirus (TD and wildtype) in peripheral blood and cardiac samples of myocarditis and DCM patients demonstrated that the higher the proportion of TD forms with deletions greater than 37 nucleotides, the lower the viral load, suggesting that, in vivo, there is interference [12,55].

If so, this is not the classical defective interfering activity as seen with DI enteroviral genomes [7], because the CVB3-TDs probably do not acquire an advantage from the wildtype genomes, as the type of positive-strand initiation is dictated by the lack of the 5′ sites essential for efficient wildtype positive-strand initiation. This inhibition of replication and translation may occur once sufficient populations of the CVB3-TD are present because of competition within the cell for host factors and ribosomes. However, the CVB3-TD are not dependent upon the replication of the wildtype virus for replication. Thus, TD viruses interfere with wildtype viruses, but are capable of replicating, unlike classical DI genomes [123].

Viral 2A protease cleavage of eIF4G in CVB3-TD51-transfected human cardiomyocytes was observed [12], but these transfections did not produce detectable VP1 by Western blot after 24 h, unlike wildtype CVB3 transfections [12]. It must be considered that the lack of detectable VP1 in the cells transfected by TD51, despite the expression of eIF4G cleavage, is probably due to the fact that each translation of a positive-strand RNA produces only one VP1 protein and 2A protein, whereas the enzymatic activity of one 2A protein will result in the cleavage of many copies of eIF4G, as noted by the authors [12]. These Western blot data demonstrate that even a very low level of CVB3-TD51 can alter the cellular environment through protease activity. It is therefore easier to comprehend how the expression of CVB3 2A protease in the hearts of transgenic mice provided evidence of cardiomyopathy [124,125]. The lower levels of 2A protease expressed by a much more slowly replicating TD virus population in a cell are likely to allow cleavage of dystrophin without immediate cytolysis of the cardiomyocyte [124,125,126]. 

### 6.3. Nonlytic Production of Virus via Exosomes May Allow Mixed Populations to Be Transferred in New Infections in Tissues

One outcome of packaging negative-strand RNA is to lower the efficiency of virions to productively infect, as nearly half of these TD virions contain, instead of a positive-strand RNA, a negative strand that is incapable of producing viral replication proteins. As it has been shown that enteroviruses may be excreted from cells without cell lysis in exosomes containing multiple viruses [127,128,129,130], the potential for maintaining a population of TD genomes is higher, as a collection of genomes can be transferred in an exosome. As the TD enterovirus population in dilated cardiomyopathic hearts may contain a minor population of wildtype enterovirus [12,53,55], exosomal transmission may provide selective advantages for maintaining both the minor wildtype and the major TD viruses in transmission [131]. A recent study has demonstrated that each exosome contains only the progeny of a single parental virus [132]; each wildtype parental virus will generate a progeny population containing both wildtype and TDs in exosomes in the restrictive environment of the differentiated cell. Over time, the TDs will dominate the cardiac infection, as seen in the mouse model of CVB3-induced heart disease, due to the high rate of TD production in cardiomyocytes over time. These results suggest that there may be a TD genome-induced interference effect with wildtype virus replication. 

## 7. Production of TDs as Part of Wildtype Infection in Some Cells: Is There a Cooperative Advantage?

In vivo, many tissues can contain cells that are not quiescent, but instead are in a phase of the cell cycle in which replication is not optimal. Production of TD enterovirus populations—following a wildtype infection—may occur for a brief period of time even in such cell populations; however, because TDs serve as effective mRNAs [12,71,74], the expression of viral proteins, particularly the viral proteases, may reach levels that sufficiently alter the cytoplasm to provide an environment for efficient wildtype replication. A part of this must be through nucleo–cytoplasmic transfer of host factors via cleavage of nucleoporins by 2A protease [70]. Evidence for this stage was provided by single-cell assays of translation using an efficient protein tag in which a phase of viral RNA replication may be unsuccessful, and may be followed by restart of translation [133]. In many cases, this second round of translation resulted in a successful round of RNA replication, which seems similar to hypothetical conditions in which normal replication is impaired, but translation of the viral RNA, and potentially TD RNA as well, can produce a suitable environment for successful replication. 

Although CVB3-TD50 has been shown to restrict the replication of wildtype CVB3 in co-transfection [12], these experiments used a mixed population with 19-fold higher CVB3-TD50 RNA than wildtype RNA. It seems likely that in cells with a high rate of translation, too few TD genomes will naturally be produced to interfere with the initial wildtype replication. There are many stages of infection of cells and tissues in human beings before sufficient virus is produced to be excreted, and most of these tissues and cells are not likely to be as permissive as the laboratory model system of HeLa cells [134]. Consequently, even very short-term expression of TD genomes in some cells in cycle arrest could give a boost to virus protein translation, which in turn may alter the cytoplasm to a more permissive condition for enterovirus replication. This altered environment may then allow the tremendous advantage of normal positive-strand initiation to produce wildtype virus genomes and not TD genomes. It is thus possible that enterovirus TDs may be, at best, a normal but very transient event in replication of wildtype viruses in many cell types.

## 8. TDs and the Immune Response

### 8.1. TDs Produce Viral Proteins When Levels of RNA Replication Are Low

As it has been shown that the level of TD virus replication is very low even in HeLa cells [76], how do these viruses manage to evade the innate immune response for long periods of time? To start with, the level of double-stranded RNA sensed by RIG-I and MDA5 [135] is low compared to wildtype virus due to the very low level of positive-strand replication. In addition, enteroviral proteases cleave many of the factors leading to innate immune responses (reviewed [136,137,138]), and although the TDs replicate poorly, they exhibit good expression of these proteases [12]. Thus, despite slow replication, TD viral RNA can still obstruct the pathways to an active innate immune response.

### 8.2. The Adaptive Immune Response That Is Very Likely to Be Induced in Humans in the Course of the Acute Infection, Does Not Clear TD Persistence

In the CVB3-induced myocarditis murine model, and in human patients with enterovirus-associated disease [139], the acute infection induces antiviral antibodies. Although immune deficiencies can lead to persistent enterovirus infections [33], enterovirus-linked heart disease has not been found specifically in that population, but seems more frequent in the general population. As TD enteroviruses replicate at low levels, it seems reasonable that the TD virus in tissues is largely present intracellularly for longer periods of time. The transmission of a replicating TD virus to new cells is likely to happen in part through extrasomal vesicles, which are largely resistant to antiviral antibodies, the primary method of enterovirus immune clearance. Nevertheless, the timing of the adaptive immune response (in mice, coinciding with clearing of the wildtype virus) suggests that early immunity or vaccination is likely to protect against enteroviral persistence. If infections occur in the context of pre-existing immunity, many sites of enterovirus pathogenesis may have a very limited infection, and consequently no significant TD generation. Vaccination is likely to prevent a high level of acute enterovirus infection in sites of the heart, central nervous system, or the pancreas in humans [140]. As vaccination has been so effective in the prevention of PV infections and poliomyelitis, this is likely the best method of decreasing the replication of other enteroviruses during acute infections. Identification of which enteroviruses pose the greatest risk of chronic disease remains a challenge.

### 8.3. Can the Presence of TD Enterovirus-Infected Cells Trigger Continuing Immune-Mediated Inflammation?

It is clear that the persistence of enteroviruses in the heart beyond the period in which the adaptive immune response to the acute infection occurs is likely to generate an inflammatory response and autoimmunity to cardiac antigens. Continuing pancreatic islet infiltration of T cells and generation of autoantibodies to pancreatic islet antigens has also been noted in patients with T1D in whom enteroviral antigens and enteroviral RNA are detected [4]. In enterovirus-associated cardiac disease, the level of interferon-β mRNA in peripheral blood correlated with higher levels of enterovirus TDs of less than 37 nucleotides [55]. This suggests that the higher viral loads of these TDs in the peripheral blood are indicative of a greater level of replication, which might induce a greater innate immune response. The presence of persisting enterovirus is not invisible to the immune system, and generation of a low-level response of both the innate and adaptive immune system may be occurring. It can be surmised, then, that there is a very slow race between TD virus replication and the killing of infected cells by the immune response, a response that occurs variably in individuals and in specific tissues. If that inflammatory environment enhances the pathogenic effects of a long-term infection, even if the TD virus loses out, the consequences of that inflammation, as in the loss of islets in the pancreas, or fibrosis and cardiomyopathy in the heart, may remain.

## 9. Treatment

Can TD enteroviruses be eliminated by antiviral or immune modulating treatment? There are inhibitors of 2A and 3C proteases and inhibitors of the replication protein 2C [141,142], as well as agents to alter capsid structure [143]. The effects of capsid-binding compounds are probably insufficient to clear TD infections, as the low yield of encapsidated virus and the use of exosomal egress suggest that this type of antiviral compound will be ineffective. In a study of enterovirus-associated DCM, treatment with interferon-beta was effective in clearing persistent virus infection [144], which offers hope, as does the recent effective treatment to clear hepatis C chronic infection [145]. In all these potential therapies, detection of patients with persistent enterovirus infections is a necessary, but not a trivial, task, being difficult in cases of T1D, cardiomyopathy, and diseases of the central nervous system due to the low-level persistence of enterovirus in disease sites that are not easily sampled; this difficulty in diagnosing persisting enterovirus infections has slowed the development of treatments. Recent work demonstrating the presence of enterovirus TD genomes in blood or plasma of myocarditis patients suggests that these easily obtained samples might provide a biomarker for enterovirus heart disease treatment at a stage where treatment might be effective [55].

## 10. Final Thoughts on TD and Enterovirus Persistence

The discovery of the existence of naturally occurring enterovirus populations with 5′ terminally deleted genomes and studies of CVB3-TD has illuminated previously dark corners of enterovirus biology, involving the induction of pathology and persistent infections. This review has raised as many questions (Table 1), as it has discussed findings regarding this mechanism of enterovirus persistence. As enterovirus infections are common, the exploration of this mechanism in the pathology of enterovirus disease is important and merits further research. It is likely that this research will result in important findings in treatment and prevention.

## Figures and Tables

**Figure 1 vaccines-10-00770-f001:**
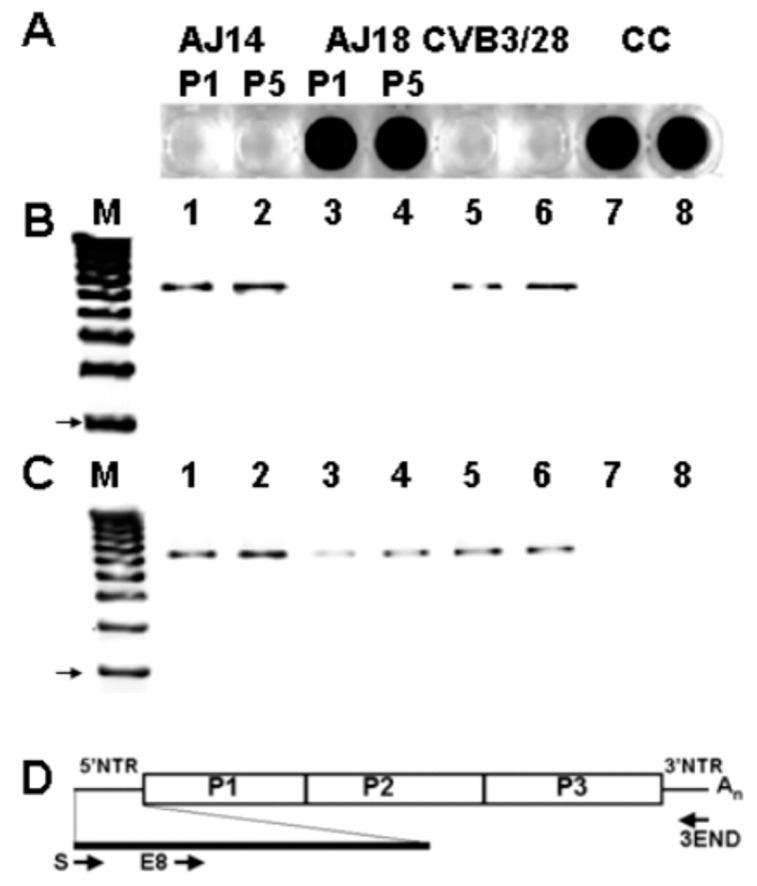
Infection of HeLa cells with homogenate of CVB3-infected AJ heart at day 18 does not induce cytopathic effect, but does replicate CVB3 RNA. (**A**) HeLa cultures inoculated with homogenates of hearts at days 14 p.i. (AJ14) and 18 p.i. (AJ18) passed once (P1) and five times (P5) in HeLa inoculated with parental virus (CVB3/28) or uninfected cell culture (CC). Cells were stained with crystal violet. (**B**,**C**) Total RNA purified from HeLa-passaged AJ14, AJ18, CVB3/28-inoculated HeLa cells, and uninfected cells was used for RT-PCR with the primer 3END and with primers S (**B**) or E8 (C). The arrow indicates a 3 kb band. Lane M, 1 kb DNA ladder; lanes 1 and 2, AJ14 P1 and P5, respectively; lanes 3 and 4, AJ18 P1 and P5, respectively; lanes 5 and 6, CVB3/28; lanes 7 and 8, uninfected cell culture. (**D**) Relative positions in the CVB3 genome of primers used. Reproduced with permission from [50] 2005, American Society for Microbiology.

**Figure 2 vaccines-10-00770-f002:**
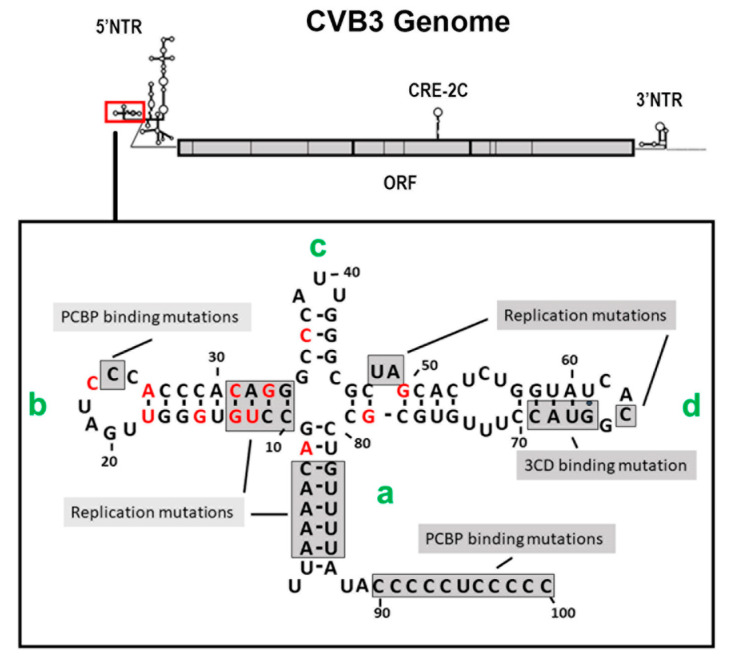
Cloverleaf of CVB3. Green letters (a, b, c, and d) refer to stem and stem-loops. Red letters indicate the first nucleotide of CVB-TDs generated in cell culture or in vivo. Shaded regions indicate sites of mutations in enterovirus genomes.

**Figure 3 vaccines-10-00770-f003:**
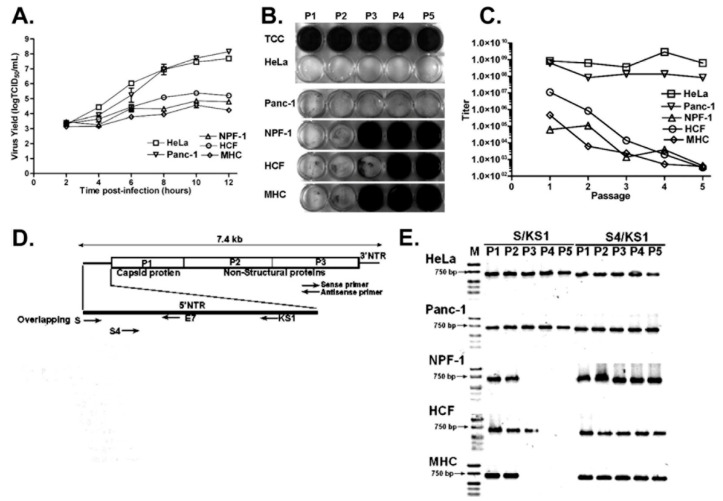
CVB3 replicates, but does not induce, CPE in the indicator HeLa cell monolayers following sequential infection of replication-restricted primary cells. (**A**) Single-step growth curves of CVB3/28 in primary cell cultures NPF-1, MHC, and HCF, and in immortal cell lines HeLa and Panc-1 (inoculated with an MOI of 20 TCID50 per cell). Error bars show standard deviations. (**B**) Supernatants of CVB3/28 passaged in primary cell cultures lose the ability to cause CPE in HeLa cell monolayers. P1 to P5, HeLa cells infected with supernatants of passages 1 to 5; TCC, uninfected HeLa cell control. Cell cultures were fixed in acetic acid/acetone and then stained with crystal violet. Titers at each passage were determined either by CPE on HeLa cell monolayers or by real-time quantitative RT-PCR. (**C**) Titers from serial passages shown in panel B are plotted. (**D**) Relative positions in the CVB3 genome of primers used in this study. (**E**) CVB3/28 RNAs in primary cell cultures detected by RT-PCR. Lane M, Hi-Lo DNA marker (Minnesota Molecular, Inc., Minneapolis, MN); lanes P1 to P5, passages 1 to 5. Arrows indicate 750 bp bands. Reprinted with permission from [61], 2008, American Society for Microbiology.

**Figure 4 vaccines-10-00770-f004:**
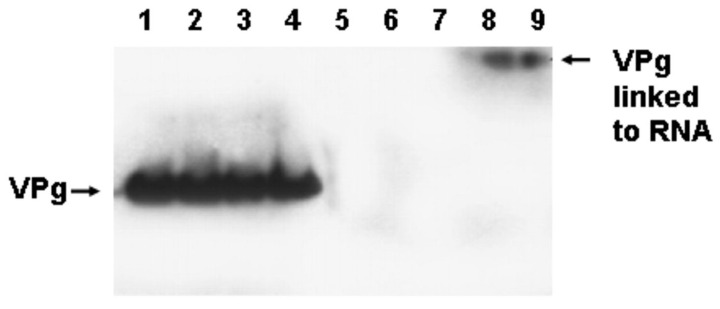
CVB3/TD viruses have VPg covalently attached to genomic RNA. Viral RNA was purified from stocks of the CVB3/28, TD8, and TD50 viruses (10^5^ TCID_50_ units) and analyzed by Western blotting. Blots were probed with an antibody to PV1 VPg (N10). Lane 1 (PV1 Sabin), lane 2, 5 (CVB3/28), lane 3, 6, 8 (CVB3/TD8), lane 4, 7, 9 (CVB3/TD50). Lanes 1–4, RNA was treated with RNase A/T_1_; lanes 5–7, RNA was treated with RNase A/T_1_ and proteinase K; lanes 8 and 9, RNA was untreated. Adapted with permission from [50] 2005, American Society for Microbiology.

**Figure 5 vaccines-10-00770-f005:**
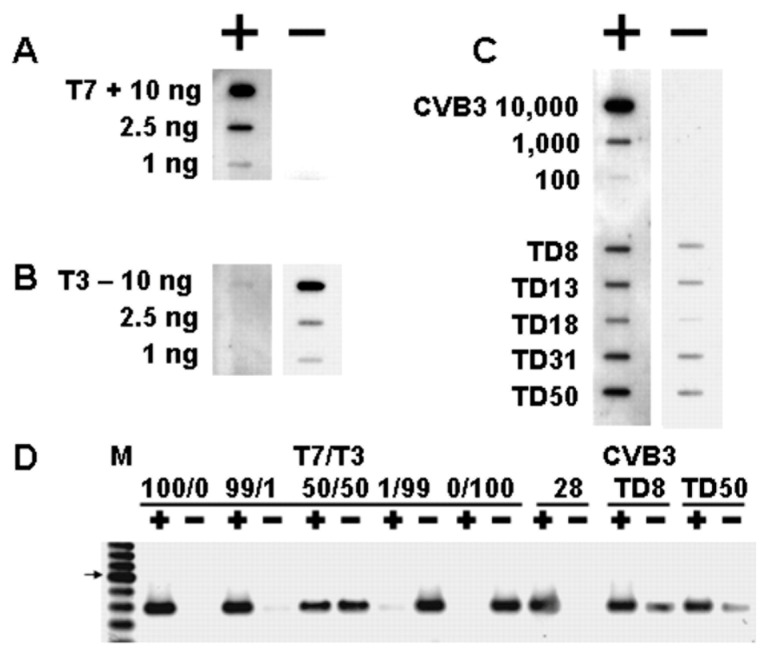
Positive and negative strands in wildtype and TD virions. (**A**–**C**) Slot blots of control and virion RNA detected with labeled strand-specific oligonucleotides. (**D**) Purified strands of control mixtures of strands or virion RNA amplified with RT-PCR and agarose electrophoresed. Reprinted with permission from [50], 2005, American Society for Microbiology.

**Figure 6 vaccines-10-00770-f006:**
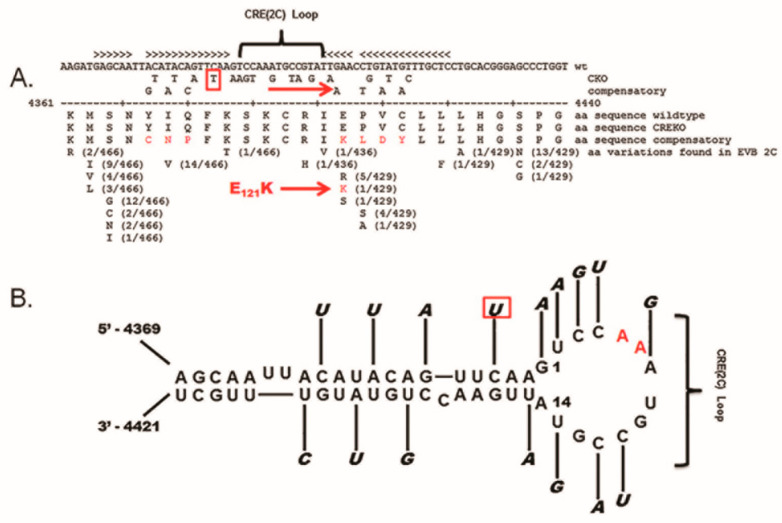
Compensatory mutations are unlikely in the stem of the CRE-2C. (**A**) Sequence analysis using NCBI was used to explore the possibility of compensatory mutations in the CRE-2C of CVB3. Potential amino acid variations resulting from compensatory mutations to restore the CRE-2C RNA structure are noted. All variations in the CRE-2C-encoding sequence that would affect the amino acid sequence of the 2C protein are noted. Only 1 amino acid variation identical to those resulting from compensatory mutations was reported to exist in 429 EV-B 2C aa sequences examined (box and arrows). (**B**) Location of this compensatory mutation in the CRE-2C secondary structure (box). Reprinted with permission [77], 2016, Elsevier.

**Table 1 vaccines-10-00770-t001:** Unresolved questions concerning TD enteroviruses.

What are the cellular conditions that generate TD genomes in differentiated cells? What difference in host environmental factors in differentiated quiescent cells prevents normal positive-strand initiation and leads to TD viral genomes? Are these conditions a result of cell cycle arrest?
What is the mechanism for nucleotidylation of VPg in TD initiation of positive-strand enterovirus RNA, and what does this mean for the functional assays of 3D polymerase and VPg? As multiply mutated CRE-2C reverts in the CVB3-TD-CKO, despite retention of the 5′ terminal deletion, what is the role of the CRE-2C in the replication of CVB3-TD?
As these TD viruses package negative- as well as positive-strand genomes, is the replication of single-stranded RNA the critical factor in packaging of enterovirus genomes, rather than an RNA-encoded signal?
This work has focused on a few enterovirus B genotypes, but the question arises whether other enterovirus types, indeed other picornavirus genomes, which utilize the same mechanism for efficient and specific virus positive-strand initiation, are prone to generation of TDs as well. If cell cycle arrest generates TDs from enterovirus B genotypes, will cell cycle arrest generate TDs from other picornavirus species?
What are the effects of TD enteroviral infection on cardiomyocytes, neuronal cells, and beta cells of TD virus infection? This question requires the ability to express TD viruses in cells with efficiency, and examine cultures with low-level infections. If host factors or cell cycle conditions are defined for the generation of TD positive-strand initiation, can these be used to generate cultures in which the majority of cells are infected with TD viruses to address the effects of TD virus infection?
How does the presence of a TD virus population majority prevent wildtype virus replication and translation at normal levels? Is this simply competition for host factors? If a population of cells with TD virus infection is generated by host factor/cell cycle conditions, can these be superinfected with wildtype virus to determine whether there is interference by the TD viruses? To what extent does the exosomal transmission of enteroviruses allow the minority wildtype population to persist?
If such a poorly replicating enterovirus variant is present in human tissues following acute infection, how common is this in enterovirus-associated human disease, given the technical requirements for detecting such a low-level infection? Given the discovery of TD enteroviruses in peripheral blood plasma of acute myocarditis patients [55], is this predictive of cardiac TD enterovirus persistence? Is the presence of TD enterovirus in plasma an indication of TD enterovirus persistence in other tissues?
What are the most important mechanisms for these variants to avoid immune clearance, and how might treatment for persistent enteroviruses utilize these targets? We know a great deal about immune evasion by wildtype virus. Do the same mechanisms allow persistent TD virus infections?

## Data Availability

Not applicable.

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
