# Peer review of "Persistent Enterovirus Infection: Little Deletions, Long Infections"

_vaccines, 2022, doi:10.3390/vaccines10050770_

Round 1

Reviewer 1 Report

The review titled “Persistent enterovirus infection: little deletions, long infections” provides a comprehensive description of the advances made in the field of enterovirus persistence. The author and her colleagues made the initial discovery that enterovirus persistence in cardiac tissue is linked to genomic deletions of the 5’ terminal extremity of the viral RNA genome. The first part of the manuscript is a complete description of persistent forms of enteroviruses focusing mainly on coxsackievirus. Their reduced replication, and the cells they can be cultured in are described. The 5’ terminal genomic deletions observed in cell culture, murine models and in patients are reviewed. The molecular consequences of these 5’ terminal deletions are described: impact on the stem loop I structure of genomic RNA and prevention of replication complex formation, while keeping the covalent linkage of viral VPg. These defects in replication lead to a reduced ratio of positive to negative RNA strands, inducing the encapsidation of negative strand RNAs in viral particles. The roles of the host protein hnRNP-C and the viral genomic region Cre are discussed at the molecular level in the context of replication of the persistent forms of RNA.

The review is well written and covers the recent advances made in the field, from the initial discovery of deleted viral genome sequences, and detection in both study models as well as patients, to the consequences of the terminal deletions on molecular mechanisms for viral replication. I have a few comments on several point that could be addressed:

  1. Line 90: It is noteworthy that enteroviruses can infect different cells in the heart. Besides myocytes, they can infect fibroblasts and endothelial vascular cells, which are not quiescent.
  2. Line 117: The Cre element is not in 2C for all enteroviruses. HRV (which is classified as an enterovirus) usually harbors this element in the genomic P1 region encoding the capsid proteins.
  3. Line 339: “ininitial” should be “initial”.
  4. Line 361: In the Bouin et al. paper, the average time between diagnosis of patient and sampling was close to 100 months. It is unlikely that the shift was not done. An issue that has not been addressed in this study is the possibility of reinfection. As the study focuses on the 5’UTR, the author failed to demonstrate if the different populations observed were originated from the same infection, or from different infections. It would be interesting to develop on the reinfection events as patient are not kept in a sterile environment as opposed to cell culture and murine models.
  5. Line 568: “Wbe” should be “be”
  6. Line 635: “detecTable” should be “detectable”
  7. Line 645: “usinfunction” should be “function”
  8. Line 906: “perpheral” should be “peripheral”

Altogether, this is a well written review that sums up the recent advances in the field. It also highlights the different facets of enterovirus persistence that are not fully understood as well as the future challenges that researchers need to overcome to understand the specific mechanism(s) leading to enterovirus persistence. This is particularly interesting as these infections usually are undetected and hard to diagnose. As infections by EV-A71 and EV-D68 are on the rise, and these viruses can infect the CNS, understanding these mechanisms could lead to the identification of new therapeutic targets. To date, the terminal deletion have not been identified in emergent enteroviruses, but these viruses share common features, and it is important for researchers to be prepares to fight potential future outbreaks.

Author Response

I appreciate the comments on this review.  Specific points are addressed as follows.

  1. Line 90: It is noteworthy that enteroviruses can infect different cells in the heart.Besides myocytes, they can infect fibroblasts and endothelial vascular cells, which arenot quiescent.

However, in the particular case of cardiac enterovirus infection in human beings and murine models, the chronic stage of enterovirus infection demonstrates signal (through in situ hybridization) in myocytes rather than fibroblasts and vascular tissue.  This was explored in the 1990s by Drs. Kandolf and Klingel using radioactive in situ hybridization which allowed longer exposures to get considerable signal (without a high level of background) demonstrating the myocyte location in chronic enterovirus infection. 

To enhance the clarity of this issue,  I have inserted a sentence at lines 171-172:

“Although studies using carrier cultures can demonstrate mutations which may adapt the virus to a particular cell type or changes in the cells in the culture in adaption to the virus [36,37], this does not seem characteristic of persistent in vivo infections of the heart or pancreas in which the tissue is well differentiated and not dividing. In particular, persistent infections within the heart demonstrate enterovirus persisting in the well differentiated, non-dividing cardiomyocytes [18,38].”

  1. Line 117: The Cre element is not in 2C for all enteroviruses. HRV (which is classifiedas an enterovirus) usually harbors this element in the genomic P1 region encoding the capsid proteins.

I have altered this sentence (lines 199-200) to clarify that point:

“The ORF also contains a cis-acting replication element (CRE-2C or OriI), a RNA structure in the region encoding a viral protein, 2C, in enteroviruses A, B, C and D [49].”

3. Line 339: “ininitial” should be “initial”. 

Corrrected.

  1. Line 361: In the Bouin et al. paper, the average time between diagnosis of patient andsampling was close to 100 months. It is unlikely that the shift was not done. An issuethat has not been addressed in this study is the possibility of reinfection. As the studyfocuses on the 5’UTR, the author failed to demonstrate if the different populationsobserved were originated from the same infection, or from different infections. It wouldbe interesting to develop on the reinfection events as patient are not kept in a sterile environment as opposed to cell culture and murine models.

There is that advantage to the murine model of persistent enterovirus infection. However, given the high proportion of human cases in which TD viruses were identified and a minor population of wildtype was present (8 of 9), it seems very unlikely that so many of these cases had multiple enterovirus infections of the heart.  In addition, in References 53 and 55, in which genotypes of the enteroviruses in clinical samples are determined, only one strain (all in the enterovirus B species) is found in each patient. Also, acute infections are likely to have a much higher level of wildtype 5’NTRs instead of the barely detectable level seen by Bouin and colleagues.  To address this comment:

“Despite having clearly shown that CVB3-TDs can replicate in mice and cell culture without any wildtype virus [50,54,61,76,77], these observations suggest the possibility that there may nonetheless be a selective advantage to having a minor population of wildtype in the cardiomyocyte enterovirus quasispecies. Alternatively, and we deem this more likely, in those tissues assayed by Bouin and colleagues, the shift in quasispecies from wildtype to TD genomes may not have been completed at time of sampling. Although a co-infection by another enterovirus during the course of the development might provide intact 5’ termini, the very low level of the wildtype 5’ termini would not be characteristic of an acute infection [50,61,63,76,77].”

The following typographic errors have been addressed.

5. Line 568: “Wbe” should be “be” : I can’t find this error.

6. Line 635: “detecTable” should be “detectable”. Corrected.

7. Line 645: “usinfunction” should be “function”. Corrected.

  1. Line 906: “perpheral” should be “peripheral”. Corrected.

Reviewer 2 Report

The author reviewed the mechanism of persistent infection of coxsackievirus B3. This is a well written, interesting, and useful contribution, which I think is entirely suitable for publication in Vaccines.

Minor points:

  1. Line 202: Please delete one “murine”.

Author Response

I thank the reviewer for the comments.

The specific typographic error on line 202 has been corrected (now line 287).

Reviewer 3 Report

1. This manuscript clearly analyzes the mechanism of how enteroviruses derive persistent infections, mainly because of the high incidence of persistent enterovirus infections due to patients' immune deficiencies, and the authors found that persistent enterovirus infections are mainly caused by mutations or deletions in the cloverleaf structure at the 5' end of the viral genome, which is called TD. Enterovirus-TD itself has a low replication rate and reduces the replication of wild-type enterovirus, but persistent infection with enterovirus-TD has a significant impact on patients with persistent myocarditis. The authors also describe the treatment of persistent enterovirus infection, in which agents that alter the capsid structure are not effective in suppressing the infection, but interferon-beta is effective in suppressing persistent enterovirus infection. The authors also state that although the viral load of enterovirus-TD is small at the disease site, the viral genome can be detected in the blood of patients with myocarditis and can be used as an indicator.

2. There is a duplication of symbols in the title 4.3 in the author's text, please correct it.

3. Is the contrast brightness of WB data in Figure 1B and C too bright, because the background of WB data is very white, is there an original image file for comparison? Please provide the raw data.

the author's text description of the wildtype is wrong whether it can be modified to wild type.

Author Response

I thank the reviewer for the comments.

  1. This manuscript clearly analyzes the mechanism of how enteroviruses derive persistentinfections, mainly because of the high incidence of persistent enterovirus infections due topatients' immune deficiencies, and the authors found that persistent enterovirus infections are mainly caused by mutations or deletions in the cloverleaf structure at the 5' end of theviral genome, which is called TD. Enterovirus-TD itself has a low replication rate andreduces the replication of wild-type enterovirus, but persistent infection with enterovirus-TD has a significant impact on patients with persistent myocarditis. The authors alsodescribe the treatment of persistent enterovirus infection, in which agents that alter thecapsid structure are not effective in suppressing the infection, but interferon-beta iseffective in suppressing persistent enterovirus infection. The authors also state thatalthough the viral load of enterovirus-TD is small at the disease site, the viral genome canbe detected in the blood of patients with myocarditis and can be used as an indicator.

I disagree with the statement that enterovirus persistent infections are mainly caused by the patients’ immune deficiencies. Immune deficiencies great enough to cause persistent infections of enteroviruses are rare and do not generate the 5’ terminal deletions characteristic in dilated cardiomyopathy and the animal models of this disease.  In this review I state: “Although immune deficiencies can lead to persistent enterovirus infections [33], enteroviral linked heart disease has not been found specifically in that population but seems more frequent in the general population” (lines   970-973). I suggested (lines 1009-1012) that treatment with antivirals that alter capsid structure might not be effective but such treatments have not been tested. 

  1. There is a duplication of symbols in the title 4.3 in the author's text, please correct it.

The heading on line 543 has been altered to 4.5.

  1. Is the contrast brightness of WB data in Figure 1B and C too bright, because thebackground of WB data is very white, is there an original image file for comparison?Please provide the raw data.

the author's text description of the wildtype is wrong whether it can be modified to wildtype.

I appreciate the point however this figure is from a publication in Journal of Virology in 2005 (used with permission).

I am not sure to what the statement “the author's text description of the wildtype is wrong whether it can be modified to wildtype” refers.  However, for this review the term “wildtype” refers to full length enterovirus genomes without 5’ terminal deletions.  I have inserted a sentence on lines 277-278:

For this review, wildtype refers to full length enteroviruses without 5’ terminal deletions.

Round 2

Reviewer 3 Report

I do not have anymore comments in this revised manuscript.